# Effect of Graphite Nanoplatelet Size and Dispersion on the Thermal and Mechanical Properties of Epoxy-Based Nanocomposites

**DOI:** 10.3390/nano13081328

**Published:** 2023-04-10

**Authors:** Elsye Agustina, Jeung Choon Goak, Suntae Lee, Yongse Kim, Sung Chul Hong, Yongho Seo, Naesung Lee

**Affiliations:** Hybrid Materials Center (HMC), Department of Nanotechnology and Advanced Materials Engineering, Sejong University, 209 Neungdong-ro, Gwangjin-gu, Seoul 05006, Republic of Korea

**Keywords:** nanocomposite, graphite nanoplatelets, thermal conductivity, tensile strength, size-dependence, dispersion, ultrasonication, attrition mill

## Abstract

This study investigated the effect of graphite nanoplatelet (GNP) size and dispersion on the thermal conductivities and tensile strengths of epoxy-based composites. GNPs of four different platelet sizes, ranging from 1.6 to 3 µm, were derived by mechanically exfoliating and breaking expanded graphite (EG) particles using high-energy bead milling and sonication. The GNPs were used as fillers at loadings of 0–10 wt%. As the GNP size and loading amount increased, the thermal conductivities of the GNP/epoxy composites increased, but their tensile strengths decreased. However, interestingly, the tensile strength reached a maximum value at the low GNP content of 0.3% and thereafter decreased, irrespective of the GNP size. Our observations of the morphologies and dispersions of the GNPs in the composites indicated that the thermal conductivity was more likely related to the size and loading number of fillers, whereas the tensile strength was more influenced by the dispersion of fillers in the matrix.

## 1. Introduction

Thermal management in advanced electronic devices has become increasingly important to extend their lifespan and performance reliability. For example, light-emitting diodes (LEDs) undergo a considerable reduction in the lifespan and quantum efficiency with increasing temperature [1], which makes heat dissipation via thermal interface materials (TIMs) and heat sinks particularly important [2,3].

Materials most frequently used for the heat sinks hitherto are metals such as aluminum or copper, which are heavy and expensive [4,5]. Therefore, there have been attempts to replace metal heat sinks with polymer composites containing thermally conductive fillers. The polymer material is attractive as the matrix of a composite due to its low weight, low cost, low thermal expansion coefficient, high chemical resistance, and easy processability [4,6]. For heat sink application, however, a polymer suffers from an extremely low thermal conductivity of 0.1–0.4 W/m·K [7]. Fillers such as alumina and silica have often been loaded to improve the thermal conductivity of polymers because they are inexpensive and lightweight. Unfortunately, because alumina and silica have low thermal conductivities of 28 and 32 W/m·K, respectively, high loading is required to dissipate the large amount of heat generated in high-power electronic devices, which in turn degrades the mechanical properties of the composite [8]. 

The current best filler candidates are nanocarbon materials, such as graphene and carbon nanotubes (CNTs), which have been receiving particular attention due to their following extremely high thermal conductivities: 2000–5000 W/m·K [9,10,11] along the in-plane direction for graphene and 2000–6000 W/m·K along the axial direction for CNTs [7]. Furthermore, they are mechanically strong, chemically inert, and light in weight. In terms of contact areas between fillers through which heat transfer occurs, however, two-dimensional (2D) graphene is a better thermally conductive filler than one-dimensional (1D) CNTs. 

Single-layer graphene is difficult to prepare; it tends to agglomerate due to strong van der Waals forces via π–π interactions, as well as roll up during processing due to its thin shape. Thus, graphite nanoplatelets (GNPs) composed of tens of graphitic layers are most frequently used as fillers for practical applications. GNP-reinforced polymer nanocomposites have shown great potential as TIMs for heat dissipation [12]. Many studies have investigated the effect of adding GNPs to epoxy matrices on the thermal conductivity of resulting nanocomposites. Yu et al. demonstrated that an epoxy nanocomposite with 25 vol% exfoliated GNPs had a thermal conductivity of 6.44 W/m·K [13]. Chandrasekaran et al. studied the influence of GNP addition to an epoxy matrix using the three-roll milling technique and observed an increase in thermal conductivity of 14% at a filler loading of 2 wt% [14]. Similarly, the thermal conductivity of GNP/epoxy nanocomposites was found to increase by 627% at a filler loading of 8 wt% compared to neat epoxy [15]. The enhancement in thermal conductivity is attributed to the thermally conductive pathways formed by the 2D structure of GNPs in the polymer matrix. As highlighted in the above studies, the thermal conductivity of GNP/epoxy nanocomposites depends on various factors such as filler size, loading level, dispersion, alignment, stiffness, and thermal contact resistance between the polymer matrix and nanofillers. While much research has focused on obtaining the high thermal conductivity of GNP/epoxy nanocomposites, the mechanical properties of these materials at high GNP loadings, which directly affect their thermal management applications, have been not much concern.

TIMs are widely used in electronic devices, but they can experience mechanical failure during thermal cycling due to a mismatch between the coefficient of thermal expansion (CTE) of the device and TIM [16]. This CTE mismatch induces strain, which generates shear and stress at the interface between them. As a result, repetitive thermal cycling can gradually cause the TIM to be damaged, reducing its thermal conductivity. Thus, the reliability of mechanical properties for TIM is important to preserve its thermal conductivity. The tensile strength of GNP/epoxy composites depends on several factors, such as the interfacial adhesion, strength of the epoxy matrix, and shape and dispersion of the fillers [17]. The dimension and dispersion of the GNP in the epoxy matrix are important for improving tensile strength, and they also play a crucial role in enhancing thermal conductivity. This is because good interfacial interaction between the fillers and the epoxy matrix allows for efficient heat transfer, which in turn can help prevent thermal cycling-induced damage to the TIM. By reinforcing the epoxy matrix with well-dispersed GNPs, both the tensile strength and thermal conductivity of the composite can be improved, reducing the risk of mechanical failure during thermal cycling.

Controlling the GNP platelet size is one of the key factors in determining the thermal conductivity of the composites in which the GNPs are incorporated. As the GNPs become larger in size, the number of contacts between them decreases, resulting in higher thermal conductivity but poorer dispersion quality; thus, the mechanical properties of the composite tend to deteriorate as the GNP size increases. As such, the GNP size should be optimized to balance the trade-off between thermal conductivity and mechanical strength. This study investigated the effect of GNP size and dispersion on the thermal conductivities and tensile properties of epoxy composites. GNPs of different sizes were derived by mechanically exfoliating EG using high-energy bead milling and sonication.

## 2. Materials and Methods

### 2.1. Materials

The raw EG used herein was purchased from Hana Chemtech (Korea). Epikote 862 (Bisphenol F; Momentive) was used as the epoxy resin, HN2200 (3 or 4-methyl-1, 2, 3, 6-tetrahydrophthalic anhydride) (Hitachi Chemicals) as the curing agent, and N,N-dimethylbezylamine (≥99%, Sigma–Aldrich) as the accelerating agent. Isopropanol (IPA) (ca. 99.5% purity, Samchun Pure Chemical) and acetone (ca. 99.5% purity, Samchun Pure Chemical) were used as solvents for the mechanical treatment and EG/epoxy composite fabrication, respectively.

### 2.2. Mechanical Treatment

EG powders were attrition-milled in IPA to reduce their particle sizes using 1-mm-diameter zirconia beads at 2000 rpm for 90 min. Thereafter, the milled EG (denoted as “M-EG”) was obtained by filtration and drying in an oven.

### 2.3. Preparation of the EG/Epoxy Composites

EG or M-EG powders were added to 80 mL of acetone, where the epoxy resin was dissolved and then exfoliated using a horn sonicator (ULH-700S Sonosmasher; Ulsso Hitech) at 497 W for 2 h to provide dispersions of S2- and MS2-GNPs, respectively. Here, “S” and “M” denote sonication and milling, respectively, and the number 2 indicates the sonication time in hours. An additional 2-h sonication was performed to produce the S4-GNP or MS4-GNP epoxy mixtures. The amount of EG or M-EG was fixed at 0.25 g in each sonication batch, while the amounts of epoxy resin, curing, and accelerating agents were adjusted to obtain the pre-designed filler contents in the GNP/epoxy composites. Sonication was carried out in a 100-mL bottle held in an ice bath while the sonication tip was immersed to the same depth as the solution surface. After sonication, the acetone was evaporated under vacuum while the mixture was kept at 60 °C. Subsequently, the curing and accelerating agents were added, and the mixture was stirred for 30 min at room temperature. The ratio between the epoxy resin, curing agent, and accelerating agent was 100:85.5:0.05. The prepared mixture was kept in a vacuum oven for 1 h at 80 °C for complete degassing and acetone evaporation. The GNP/epoxy mixture was transferred to a stainless-steel mould and hot-pressed at a pressure of 10 MPa during curing for 1 h at 80 °C and then for 2 h at 120 °C. Mechanical pressing was needed to prepare the GNP/epoxy composites with filler contents higher than 5 wt% because they had extremely high viscosities. 

### 2.4. Characterisation

The morphologies and sizes of the EG particles and GNPs were observed using field-emission scanning electron microscopy (FE-SEM; model S-4700; Hitachi). SEM samples were prepared by filtering the diluted solutions of EG powders and GNPs through Pt-coated polycarbonate membranes so that the particles or platelets were individually scattered for clear observation. Atomic force microscopy (AFM; model XE-100; Park Systems) in the contact mode was used to examine the thickness of the GNPs deposited on a silicon wafer. An AFM cantilever tip having a radius less than 50 nm was operated at a force constant of ca. 7.4 N/m and at the resonance frequency of ca. 160 kHz. The AFM samples were prepared by filtering dispersions of the GNPs in IPA through polycarbonate membranes, followed by transferring the GNPs onto silicon wafers by pressing the membranes with a roller. The particle size analysis was performed on acetone dispersions of the EG and GNP powders using laser granulometry (Mastersizer 3000; Malvern Instruments). Their specific surface areas were measured by the Brunauer–Emmett–Teller method (BET; BELSORP-mini II; Microtrac) in nitrogen gas at 77 K. Crystallinity and structural defects of the EG and GNPs were characterized using Raman spectroscopy with a laser wavelength of 633 nm (Renishaw System 3000; Renishaw). The oxidation temperatures of both GNPs and composite samples were measured by thermogravimetric analysis (TGA; model TGA-50H; Shimadzu). The dispersion states of the fillers in the epoxy matrix were examined using light-transmission optical microscopy (model BX41M; Olympus). The through-plane thermal diffusivities (*α*, mm^2^/s) of circular-shaped pellets of the GNP/epoxy composites (thickness: 1 mm; diameter: 12.7 mm) were measured by a laser flash system (Flashline 3050 System; TA Instruments) at room temperature with a laser power of 500 W. The thermal conductivity (*k*, W/m·K) was calculated using the following equation, $k=\alpha \times \rho \times C_{P}$, where the apparent densities (*ρ*, kg/m^3^) of the samples were measured by dividing the mass over the volume, and the specific heat capacity (*C_p_*, J/kg K) was measured by DSC (Q200, TA instruments). Tensile tests were conducted using a tensile machine (model AGS-X; Shimadzu) at a speed of 1 mm/min. Tensile testing specimens having a 1-mm-thick dog-bone shape were prepared according to the standard ASTM-D1708.

## 3. Results and Discussion

EG is usually prepared by intercalating natural graphite flakes with strong acids or oxidizing agents, followed by high-temperature shock treatment [18]. EG can be easily exfoliated to produce GNPs due to the weak van der Waals interactions between the widened graphitic layers [19]. In this study, GNPs were produced by mechanical milling and sonication of EG. During the mechanical milling, EG particles experience an impact force via bead collision and a shear force via bead sliding [20]; the former force may cause fragmentation of EG particles, and the latter force may induce exfoliation along their graphitic layers, thus effectively reducing the planar size as well as the thickness of the EG particles [21]. During sonication, EG particles are exfoliated to GNPs by turbulent flow and micro-jets from acoustic cavitation (i.e., the formation, growth, and implosive collapse of bubbles) [22].

The size distribution of EG particles subjected to exfoliation processes was analyzed using SEM images and laser granulometry. Figure 1 shows SEM images of the raw EG, M-EG, and four GNPs (S2-, S4-, MS2-, and MS4-GNPs). Individual particles or platelets were observed by diluting and filtering the dispersed solutions of EG powders and GNPs through Pt-coated polycarbonate membranes. Platelet sizes of GNPs were measured for at least 150 platelets using SEM. On the other hand, laser granulometry was also used to measure particle sizes of the raw EG, M-EG, and four GNPs (see Appendix A). For the S2- and S4-GNPs given in Figure 1b,c, the average platelet sizes drastically decreased to ca. 3.0 and 2.0 μm, respectively. For the MS2- and MS4-GNPs, prepared by milling for 1.5 h and sonicating for 2 and 4 h, the average platelet size decreased to ca. 2.0 and 1.6 μm, respectively (Figure 1e,f). Referring to the platelet sizes given in Appendix A, the MS2-GNPs (2.0 ± 1.2 μm) showed smaller standard deviations than the S4-GNPs (2.0 ± 1.6 μm), although they had the same average size. This indicated that attrition milling was effective in preparing GNPs with a narrow size distribution. The specific surface areas of GNPs, measured using the BET method, are summarised in Appendix A. The specific surface areas of the GNPs expectedly increased with decreasing platelet size.

In laser granulometry, to observe particle size distribution, the D50 value represents the mass median diameter of all the particles. The raw EGs had a D50 of ca. 143.0 µm and a full-width at half-maximum (FWHM) of ca. 387.8 µm (Appendix A). Łoś et al. reported that sonication alone of EGs for a short period of time was inadequate to achieve a homogenous size distribution of GNPs because the fragmentation speed slowed down with irradiation time [23]. Our result also showed that the S2-GNPs, prepared by sonication for 2 h, had an inhomogeneous size distribution, as evidenced by the broad curve having a hump on the left side in the particle size distribution (Appendix A). For the S2-GNPs, the D50 was shifted to ca. 31.8 µm, and the FWHM was narrower at 136.5 µm. Increasing the sonication time to 4 h (S4-GNPs) moved the D50 value to a still smaller size of ca. 11.4 µm and resulted in a narrower FWHM of ca. 23.9 µm by downsizing the large particles corresponding to the right-most peak of the S2-GNP distribution (Appendix A).

Similarly, attrition milling solely was not effective in reducing graphite flake size [24]. Appendix A gives the average particle size of the EG, measured using laser granulometry, as a function of attrition-milling time. The particle size drastically decreased from ca. 143 µm for the raw EG (0 h) to the saturated value of ca. 43 µm after 1.5 h of milling time; this was thus chosen as the optimum milling time. For the M-EG milled for 1.5 h, the D50 was ca. 43.0 µm with a narrow FWHM of ca. 55.3 µm (Appendix A). Interestingly, the M-EG had a much narrower FWHM than the S2-GNPs.

This study combined high-energy attrition milling with sonication to produce GNPs with a small size and narrow distribution. Sonicating M-EG for 2 h, forming MS2-GNPs, shifted the D50 to a smaller size of ca. 20.3 µm and decreased the FWHM to ca. 51.8 µm. MS4-GNPs, produced by sonicating M-EG for 4 h, showed a further D50 shift to a smaller size of ca. 9.6 µm with a narrower FWHM of ca. 24.1 µm. Thus, a combination of milling and sonication led to more symmetric peaks in the particle size distribution, which indicated a small, homogeneous, and narrow size distribution of the GNPs.

The platelet thickness of each GNP was measured using AFM. Averaging the line profiles in Figure 2, the S2- and MS2-GNPs had platelet thicknesses of ca. 140 and 98 nm, respectively; the S4- and MS4-GNPs had smaller platelet thicknesses of ca. 35 and 17 nm, respectively. The data of Appendix A show the average thicknesses of S2-, S4-, MS2-, and MS4-GNPs from the measurements of at least 10 platelets for each sample. Attrition milling prior to sonication was effective not only in reducing platelet size but also in exfoliating layers of GNPs. Smaller graphite flakes can be exfoliated with greater ease than larger ones due to reduced van der Waals forces between their graphitic layers [21]. The number of graphene layers of a platelet was calculated from its thickness measured by AFM, using the approximation reported by Gupta et al. [25] that the distance between adjacent graphene layers is 0.35 nm. Thus, using the average thicknesses of ca. 33.4 and 21.3 nm given in Appendix A, the S4- and MS4-GNPs were composed of ca. 95 and 65 graphene layers, respectively. To minimize the surface energy, thin GNPs readily roll up into nanoscrolls after exfoliation, which can drastically decrease their aspect ratios and reduce the matrix–filler, and filler–filler interactions in the composite, hindering heat transfer between them [26,27]. According to Viculis et al. [28], GNPs thicker than ca. 20 nm would be favorable to maintaining the 2D platelet-like shape without rolling up.

The raw EG, M-EG, and four GNPs were characterized using Raman spectroscopy (Figure 3a). Their crystalline qualities were assessed by their G-to-D peak ratios, I_G_/I_D_, where the D-band peak at ca. 1350 cm^−1^ is caused by structural defects and grain boundaries, and the G-band peak at ca. 1580 cm^−1^ corresponds to the planar sp^2^ configuration of carbon atoms in the graphitic structure [29,30]. A decrease in the I_G_/I_D_ value then indicates an increased defect and grain boundary density in the graphitic structure. The I_G_/I_D_ ratio is also related to the particle size of GNPs, as Tuinstra et al. [31] previously reported; in that work, more intense D peaks were attributed to smaller flake sizes. The raw EG and M-EG had I_G_/I_D_ ratios of 8.0 and 7.0, respectively. For the GNPs, the I_G_/I_D_ ratios decreased in the order of S2- > MS2- > S4- > MS4-GNPs, which is in agreement with the order of their particle sizes, as shown in Appendix A. The thermal stabilities of the EG and GNPs were measured using TGA in flowing air, and their derivative thermogravimetric (DTG) curves are given in Figure 3b. The DTG peak gives the rate of mass loss (dm/dT) at a maximal temperature, which can be used to clearly distinguish the difference in the thermal behavior of materials according to their composition. The oxidation temperature, T_ox_, is defined as the peak temperature of the DTG curve and represents the thermal stability of a carbon material [32]. The T_ox_ value was 743.9 °C for the raw EG and decreased with attrition milling and longer sonication time. For the GNPs, the T_ox_ values decreased in the same order as their particle sizes and decreased I_G_/I_D_ ratios, as presented in Appendix A. The T_ox_ is a measure of the resistance to oxidation and is related to the defects and platelet sizes of the GNPs [5].

The dispersion states of the GNPs in the epoxy matrix were observed using transmission optical microscopy. We exploited the light transparency of the epoxy resin, which enables clear visualization of GNP agglomeration, dispersion, and network state at the macroscopic level without interference from the epoxy matrix, as shown in Figure 4 [33]. GNP/epoxy composite films, as thin as 1.4 µm, were prepared by pressing and curing them between two glass plates. Small amounts of GNPs, 0.3 and 1 wt%, were loaded into the epoxy so that the GNPs did not overlap in the thickness direction of the films. The SEM images of Figure 1 show the individual GNPs that were used to measure their average platelet sizes. The platelets appear agglomerated in the optical microscopy images observed at a low magnification (Figure 4). Large aggregates and small particles coexisted in the S2-GNP/epoxy composite having the 0.3 wt% loading of the GNPs (Figure 4a). As the sonication time was extended to 4 h, the large aggregates were broken down but remained with reduced sizes in the S4-GNP sample (Figure 4b). Sonication alone, even for a long time of 4 h, was insufficient to completely break down the large aggregates. The MS2-GNPs, which had been attrition-milled for 1.5 h and then sonicated for 2 h, showed a decrease in the aggregate size but an increase in their number (Figure 4c) compared with the S2- and S4-GNPs. After 4 h of sonication, the MS4-GNPs exhibited a homogeneous dispersion of small GNPs of uniform size (Figure 4d). Increasing the loading amount of the GNPs to 1 wt% (Figure 4e–h) resulted in larger agglomerates of particles, and their populations also increased for each of the four GNPs.

Thermal conductivities were measured for the GNP/epoxy composites loaded with 0–10 wt% of the four different GNPs (Figure 5a). In each case, the thermal conductivity linearly increased with a loading amount. At 10 wt% loading, S2-GNPs provided the highest thermal conductivity of 1.74 W/m·K, which was ca. 8 times higher than that of neat epoxy (0.22 W/m·K). At GNP concentrations greater than ca. 1 wt%, the thermal conductivities decreased in the order of S2- > S4- > MS2- > MS4-GNPs. In the high-loading region over 1 wt%, the thermal conductivities of the composites increased with increasing platelet size. The thermal conductivities of the composites having the same loading amounts of GNPs are affected by many factors of GNPs, including their platelet sizes, thicknesses, dispersion states, crystallinities, and contact resistances [34,35]. The physical connectivity between the GNPs within the epoxy matrix, which is determined by their loading amounts and morphological properties such as size, thickness, and dispersion, would make the greatest contribution to the thermal conductivities of the composites. The connectivity of the GNPs increases with the loading amount. Once they are connected within the epoxy in the high-loading region, the thermal conductivities of the composites increase as the number of physical contacts between the GNPs decreases, i.e., as the GNPs increase in platelet size. It is thus noted that the thermal conductivities increased with the platelet size of GNPs in the high-loading region (Figure 5a). This trend, however, was not followed in the low-loading region below 1 wt%. The thermal conductivities were higher at 0.3 wt% in the order of MS4- > MS2- > S4- > S2-GNPs, which was the reverse of their order in the high-loading region (Figure 5a, inset). The thermal conductivities increased with decreasing platelet size of GNPs in the low-loading region. Notably, the 0.3-wt% loading of GNPs increased the thermal conductivity of the composites relative to the neat epoxy for the S4-, MS2- and MS4-GNPs while was not of help in the case of the S2-GNPs (Figure 5a, inset). The order of the thermal conductivities of the composites was exactly reversed for the low- and high-loading regions. At the low-loading level of 0.3 wt%, GNPs were not physically connected with each other over a long distance (Figure 4a–d). It may appear that the smaller GNPs increased their connectivity over short distances to establish more heat-conducting channels.

Hypothetical explanation of the thermal conduction mechanism of the GNP/epoxy composites having the GNPs at a low loading of ca. 0.3 wt% is suggested in Figure 6, based on the following observation of Figure 4: the GNPs are agglomerated locally but are not connected over a long distance. The epoxy matrix behaves as the bottleneck slowing down cumulative phonon transport in the composites due to its nature of poor thermal conduction, while the GNP aggregates spread over the matrix accelerate the phonon transport in the composites. Red lines in Figure 6 represent thermal pathways in the composites, where the dotted lines illustrate slow heat conduction through the epoxy matrix, and the solid lines display fast heat conduction through the aggregated GNP fillers. As their sizes decrease from S2-GNPs to S4-, MS2- and MS4-GNPs, the distance between the GNP aggregates decreases, enhancing their connectivity over a short distance and their resultant thermal conductivities. With higher loading of GNPs, however, the GNP aggregates are in physical contact with each other, and then the other factors, such as the number of contacts between GNPs, will have greater influences on the phonon transport through the composites. Under such a circumstance, the GNPs with a larger size would have an advantage over the smaller-sized GNPs in terms of thermal conductivity.

Thermal stability of the GNP/epoxy composites having 10 wt% GNPs was examined using TGA and DTG curves, as given in Appendix A and Figure 7, respectively. The degradation temperature of epoxy material, Td, represents the maximum weight loss rate in the DTG curve [36]. The neat epoxy showed a T_d_ of ca. 398 °C. The composites filled with S2-, S4-, and MS2-GNPs had T_d_ of ca. 399, 400, and 399 °C, respectively. Loading S2-, S4-, and MS2-GNPs hardly enhanced the thermal stability of the composites. Significant improvement, however, was achieved by loading MS4-GNPs, whose composite showed T_d_ of ca. 410 °C, which was 12 °C higher than that of the neat epoxy. The higher thermal stability of the composite filled with MS4-GNPs is related to more interfacial interaction between fillers and matrix due to their larger surface area [37].

Ultimate tensile strengths were measured for the GNP/epoxy composites with the four different GNPs loaded between 0 and 10 wt% (Figure 5b). In each case, the tensile strength reached a maximum of 0.3 wt% and then decreased continuously as the loading amount increased. The highest tensile strength was ca. 60 MPa for MS4-GNPs at 0.3 wt% loadings, which was improved by ca. 56% over that of neat epoxy. While other studies [34,38,39] reported that the incorporation of GNPs into polymers had a negative effect on strengthening due to their aggregation or poor dispersion, our study showed a considerable reinforcing effect of GNPs on mechanical strength when they were loaded below ca. 1 wt%. Their reinforcing effect varied according to the type and amount of GNPs. However, loading excessive amounts of GNPs into the epoxy decreased the strengths of the composites to values even below that of neat epoxy. For example, the composites containing more than 2 wt% of S2-GNPs, or more than 3 wt% of other GNPs, had lower strengths than the neat epoxy. At such high concentrations, it is likely that the GNPs were agglomerated in the composites and were easily disintegrated during the tensile test. Agglomeration of GNPs may cause voids and cracks when the polymer fails to penetrate the aggregates [40]. Such aggregated GNPs would act as structural defects rather than reinforcing fillers [41]. Fillers generally strengthen composites because they retard crack propagation through the matrix by energy absorption, crack pinning, or crack deflection [38,42]. The fillers should be fully disintegrated, evenly distributed in the matrix, and surrounded tightly by epoxy so that they can act as effective crack retardants. Figure 5a,b shows that the thermal conductivity increased while the mechanical strength decreased with increasing GNP loading. With increased loading, GNPs became more connected, which increased the number of thermal conduction pathways; however, these contact points also acted as mechanically weak points. Thus, the loading amount should be optimized to balance the trade-off between thermal conductivity and mechanical strength.

Over the range of concentrations between 0 and 10 wt%, the strength of the composite followed the order of MS4- > S4- ≈ MS2- > S2-GNPs, i.e., increasing with decreasing GNP size. The strength is mainly determined by the interfacial areas between the filler particles and the matrix where stress transfer occurs [6]. Mechanical milling and extended sonication break and exfoliate GNPs, eventually producing GNPs of smaller sizes and thicknesses, and larger surface areas, as given in Appendix A. The MS4-GNPs, which were the smallest in size and thickness and the largest in surface area, consistently showed higher tensile strengths than the other GNPs. Our results agreed with those of Khan et al., who varied the GNP size by centrifugation and reported a greater reinforcement of smaller graphene flakes in the polymer matrix [43]. Additionally, thick GNPs may adversely affect the strength of composites due to the easy shearing and gliding of graphitic layers along the (001) basal plane [44].

There are additional reasons why small-sized GNPs are favored as fillers in composites. First, small GNPs may reduce the surface asperities that prevent tight surface contacts between a heat sink and a heat generator. In particular, thermal interface materials prefer small GNPs to produce unruffled composite surfaces [45]. Second, small GNPs may decrease the viscosities of composites and mitigate issues related to their anisotropic properties, thus facilitating manufacturing.

## 4. Conclusions

Composites were fabricated by incorporating GNPs of different sizes into an epoxy matrix. Combined attrition milling and sonication of raw EG with a large size of ca. 280 μm produced GNPs having small platelet sizes between 1.6 and 3.0 μm, i.e., S2-, S4, MS2- and MS4-GNPs. The I_G_/I_D_ crystallinities and TGA oxidation temperatures of the GNPs decreased with attrition milling and longer sonication time, most likely due to the generation of more defects as the platelet size became smaller. The thermal conductivities of the composites linearly increased as the loading amount of the GNPs increased from 0 to 10 wt%, while the tensile strength reached a maximum at 0.3 wt% and then decreased continuously with increasing GNP amount. As the size of the GNPs decreased, the thermal conductivity decreased at high loadings over 1 wt%, but this trend was reversed at the low loading of 0.3 wt%. This behavior was attributed to changes in the GNP-to-GNP connections with a loading amount. The tensile strength increased as the size of the GNPs decreased. Thus, a higher loading amount and larger GNP size increased the thermal conductivity but decreased tensile strength. Thermal conductivity was affected significantly by the connectivity and contact number of GNPs, whereas the tensile strength was strongly related to the dispersion and filler–matrix interfacial area. The loading amount and size of the GNPs should be optimized for specific applications such as TIM and heat sink, where the trade-off relationship between the thermal conductivity and mechanical strength of the composites is important.

## Figures and Tables

**Figure 1 nanomaterials-13-01328-f001:**
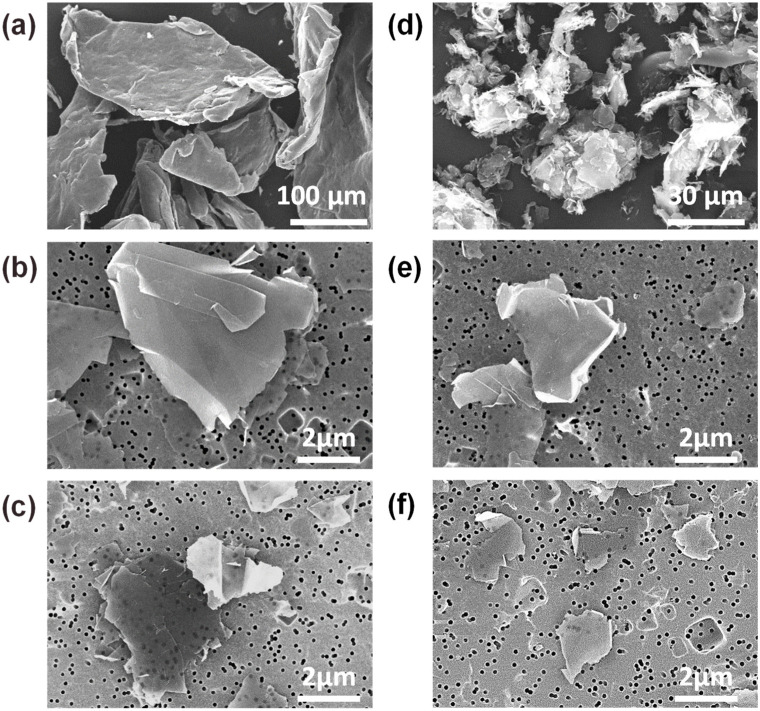
Scanning electron microscopy (SEM) images of (**a**) raw EG, (**b**) S2-GNPs, (**c**) S4-GNPs, (**d**) M-EG, (**e**) MS2-GNPs and (**f**) MS4-GNPs, where “S” and “M” denote sonication and milling, respectively, and the number indicates the sonication time in hours. EG: expanded graphite; GNP: graphite nanoplatelet.

**Figure 2 nanomaterials-13-01328-f002:**
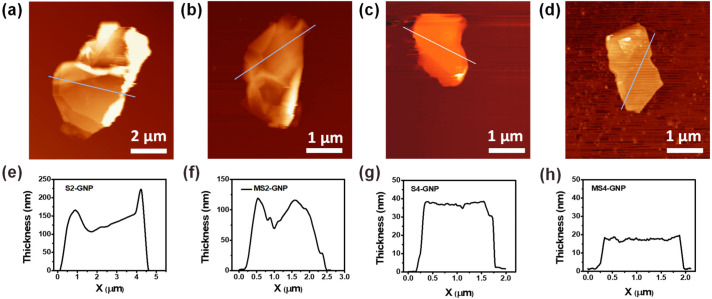
Atomic force microscopy (AFM) images and line profiles along the white lines of (**a,e**) S2-, (**b,f**) MS2-, (**c,g**) S4- and (**d,h**) MS4-GNPs, which had average thicknesses of 140, 98, 35, and 17 nm, respectively.

**Figure 3 nanomaterials-13-01328-f003:**
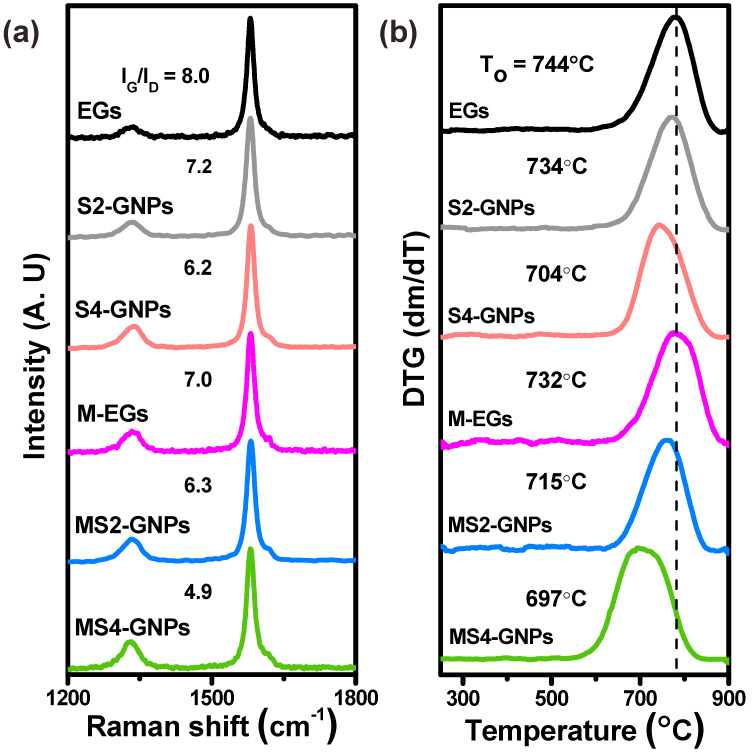
(**a**) Raman spectra and (**b**) derivative thermogravimetric (DTG) curves of EGs and GNPs. The G−to−D peak intensity ratios (I_G_/I_D_) of the Raman spectra and the oxidation temperatures (T_ox_) from the peaks of DTG curves are shown.

**Figure 4 nanomaterials-13-01328-f004:**
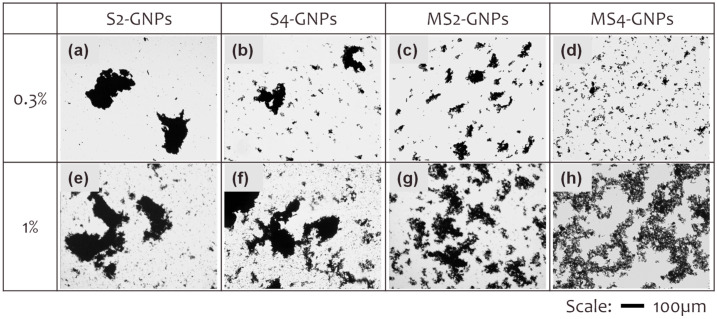
Transmission optical microscopy images of (**a**,**e**) S2-, (**b**,**f**) S4-, (**c**,**g**) MS2- and (**d**,**h**) MS4-GNPs in epoxy composites. The (**a**–**d**) and (**e**–**h**) contained 0.3 and 1 wt% of GNP fillers in the composites, respectively. GNP/epoxy composite films as thin as 1.4 µm were prepared by pressing and curing them between two glass plates to observe the dispersion states of the GNPs using optical microscopy. Scale bars correspond to 100 µm.

**Figure 5 nanomaterials-13-01328-f005:**
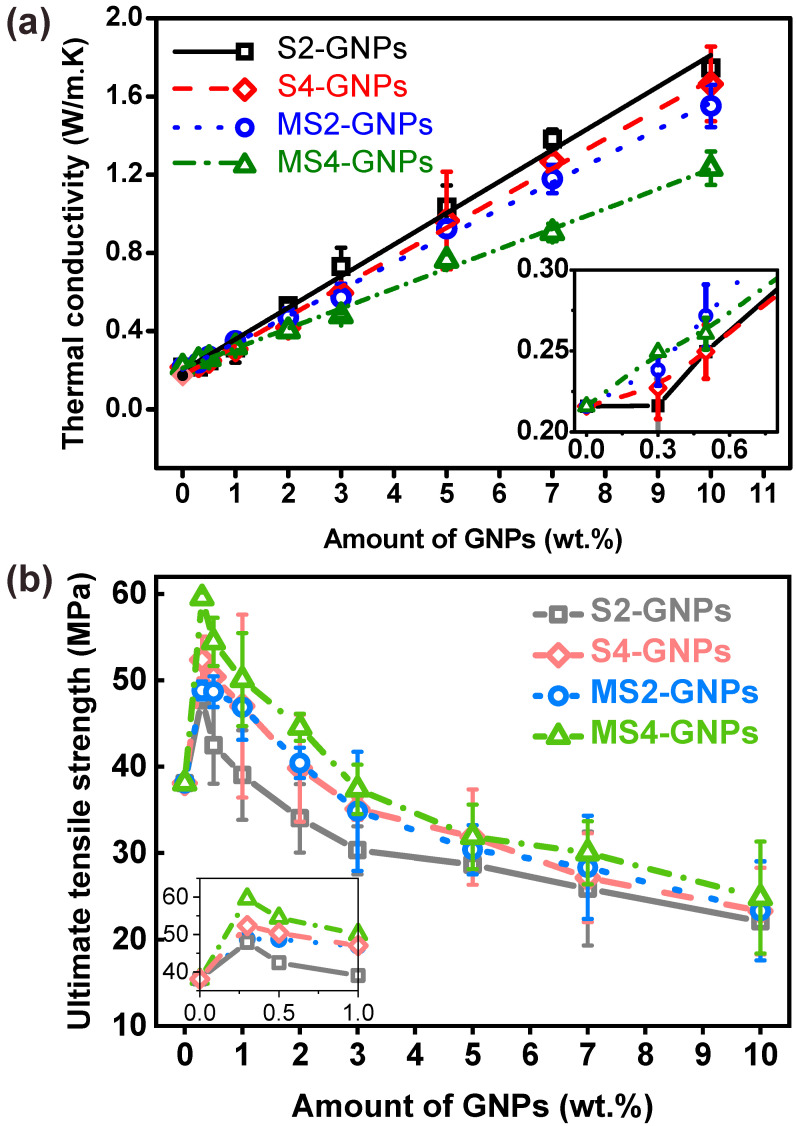
(**a**) Thermal conductivities and (**b**) ultimate tensile strengths of composites with varying amounts of S2-, S4-, MS2-, and MS4-GNP fillers. The inset in (**a**,**b**) shows the magnified graphs at low GNP concentrations.

**Figure 6 nanomaterials-13-01328-f006:**
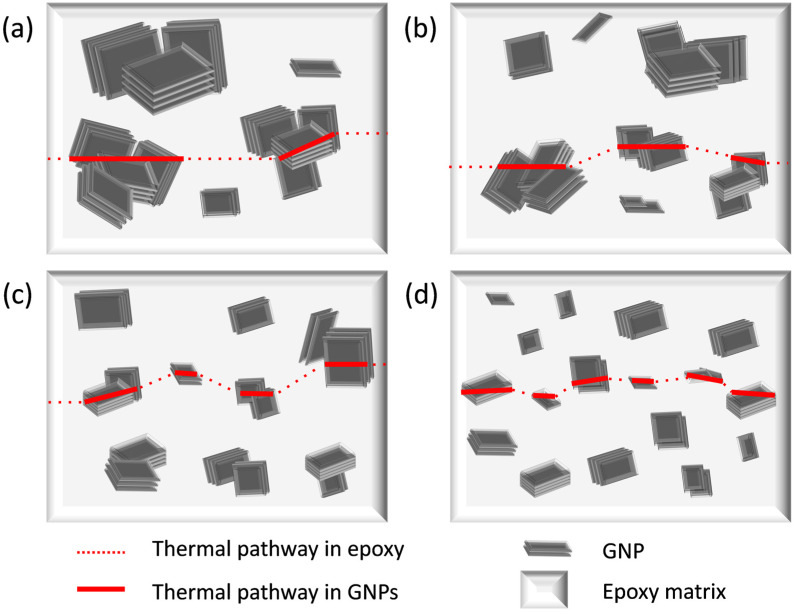
Pathways of thermal conduction for the composites loaded with a low concentration (ca. 0.3 wt.%) of (**a**) S2-, (**b**) S4-, (**c**) MS2-, and (**d**) MS4-GNPs.

**Figure 7 nanomaterials-13-01328-f007:**
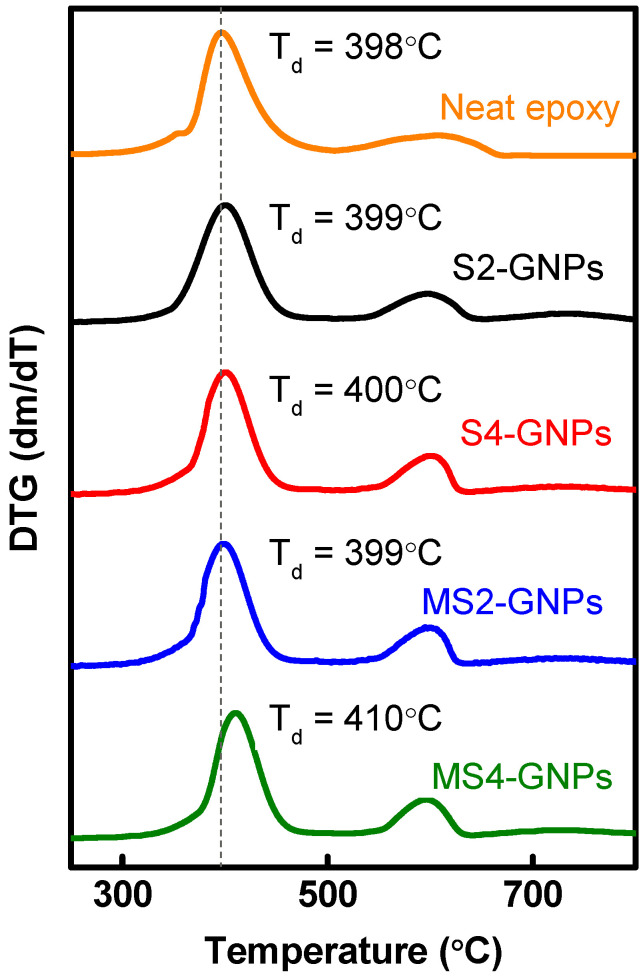
DTG curves of neat epoxy and GNP/epoxy composites loaded with 10 wt.% of S2-, S4, MS2-, and MS4-GNPs.

## Data Availability

The data presented in this study are available on request from the corresponding author.

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
