# Peer review of "Effect of Graphite Nanoplatelet Size and Dispersion on the Thermal and Mechanical Properties of Epoxy-Based Nanocomposites"

_nanomaterials, 2023, doi:10.3390/nano13081328_

Round 1
Reviewer 1 Report
The manuscript entitled ''Effect of graphite nanoplatelet size and dispersion on the thermal and mechanical properties of epoxy-based nanocomposites''adds some interesting results to literature, which would be interesting to the readers of nanomaterials journal. I would suggest publication after major revisions:
- The introduction is very brief and fails to inform the reader of the prior art. The authors should include in a couple of paragraphs at least, the most important references and their key results in terms of epoxy/GNP systems, thermal conductivity and mechanical properties. There are many reports in literature about thermal interface materials with epoxy/graphene. The number of references in the references list must increase.
- The characterisation part of the flakes produced is quite extensive which is good.
- It is not clear if these developed GNP/epoxy systems are designed as thermal interface materials, heat sinks or both. In any case, the authors should justify why the ultimate tensile strength property matters and is worth investigating and reporting in conjuction with thermal conductivity? Are structural parts in application (preparation or use phases) under tensile stress? why for example aren't the Young's modulus or elongation at break reported?
- The term ''graphite nanoplatelets (GNPs)'' is abbreviated in line 49 and used ever after which is ok. However, the term ''graphene'' is also used in the manuscript. The authors should check carefully in the literature the nomenclature and make corrections if necessary. The terms ''graphite platelets'' seems to be far from the term ''graphene''. Should graphene nanoplateles (GNPs) be used instead?
- from line 301 TGA results are discussed, which I suppose is Thermogravimetric Analysis, however Fig.7 shows ''DTG''. I would expect to see a TGA graphs showing the weight loss during a heat ramp from RT up to e.g. 800oC. I am not sure what DTG or DTGA are; they are not explained in the manuscript. Please add and discuss the TGA results.
- Could the authors comment on their selection to use Transmission optical microscopy method to check the dispersion quality and not SEM? Has the optical method been used before to study dispersion in literature?
- In section 4. Conclusions, delete lines from the journal's template: ''This section is not mandatory but can be added to the manuscript if the discussion is unusually long or complex.''
Reviewer 2 Report
In this paper, the GNPs with different size were prepared by high energy bead milling and sonication. And the effect of the size and dispersion of GNP on the thermal and mechanical properties of epoxy-based materials were systematic researched. The results were meaningful and valuable. However, before published, some defects should be modified.
1、 In introduction section, the research work focuses on epoxy based composite materials should be reviewed to explain the feasibility and necessity of GNP as an epoxy filler.
2、 The thermal stability should use the onset weightlessness temperature which was the onset temperature of TGA curves not the peak temperature of DTG curves.
3、 In fig 5. (b), the tensile strength of the samples in lower GNPs loading should be highlighted.
4、 Thermal conductivity is the product of a material's heat capacity, thermal diffusion coefficient, and density. The author just measures the thermal diffusion coefficient using the laser flash system. What`s the thermal conductivities of materials in Fig. 5(a)?
